# Development and Immunogenicity Study of Subunit Vaccines Based on Spike Proteins of Porcine Epidemic Diarrhea Virus and Porcine Transmissible Gastroenteritis Virus

**DOI:** 10.3390/vetsci12020106

**Published:** 2025-02-01

**Authors:** Mingguo Xu, Zhonglian Yang, Ningning Yang, Honghuan Li, Hailong Ma, Jihai Yi, Huilin Hou, Fangfang Han, Zhongchen Ma, Chuangfu Chen

**Affiliations:** 1College of Animal Science and Technology, Shihezi University, Shihezi 832000, China; xumingguo@xjshzu.com (M.X.); yangzhonglian@xjshzu.com (Z.Y.); lhh121004@126.com (H.L.); 15899292491@163.com (J.Y.); houhuilin11@163.com (H.H.); 2College of Animal Science and Technology, Xinyang Agriculture and Forestry University, Xinyang 464000, China; 2024200008@xyafu.edu.cn; 3Department of Biotechnology, Linxia Modern Career Academy, Linxia 731100, China; mahailong1@xjshzu.com; 4The College of Animal Science and Veterinary Medicine, Henan Agricultural University, Zhengzhou 450002, China; 13526485257@sina.cn

**Keywords:** PEDV, TGEV, S1 protein, subunit vaccines

## Abstract

Porcine epidemic diarrhea virus (PEDV) and transmissible gastroenteritis virus (TGEV) are critical viral pathogens that pose a severe threat to swine health, resulting in significant economic losses for the global pig farming industry. Vaccination remains the most effective strategy for disease prevention; however, current vaccines face challenges, including suboptimal safety profiles and inadequate protective efficacy. Consequently, there is an urgent need for the development of a safe and effective vaccine capable of providing concurrent protection against both PEDV and TGEV infections. In this study, we developed novel subunit vaccines incorporating the PEDV S1, TGEV S1, PEDV S1-TGEV S1, and PEDV S1 + TGEV S1 proteins. The immunogenicity of these vaccines was preliminarily assessed in a mouse model, suggesting a promising approach for preventing both PEDV and TGEV infections.

## 1. Introduction

Porcine epidemic diarrhea virus (PEDV) and porcine transmissible gastroenteritis virus (TGEV) are both members of the family *Coronaviridae* and the genus Coronavirus [1,2]. Based on their preferred replication sites in the intestine, PEDV and TGEV can be classified as type I viruses that specifically infect villous enterocytes [3,4]. A total of 127 porcine samples from 48 farms across six provinces in China were analyzed, revealing a PEDV detection rate of 43.0% and a co-infection rate of 12.0% between PEDV and TGEV [5]. Numerous studies have demonstrated that co-infection with PEDV and TGEV is prevalent, and may facilitate recombination between the two viruses [6,7,8,9]. Furthermore, evidence suggests that co-infection with these enteric viruses can lead to synergistic or additive effects, resulting in more extensive villous atrophy and more severe, prolonged diarrhea throughout the entire intestine [3]. Both PEDV and TGEV infections are highly contagious viral diseases that can affect pigs of all ages. Among these, the morbidity and mortality rates in suckling piglets are particularly high, leading to significant economic losses for the global pig industry. Currently, there are no effective antiviral drugs available for these two viral diseases, underscoring the critical and urgent need for the development of safe and effective vaccines.

A metagenomic analysis conducted on diarrhea and healthy samples from China revealed that 78% of the diarrhea samples contained porcine coronaviruses, while only approximately 7% of the healthy samples exhibited the presence of coronaviruses. The finding underscores potential relevance of coronaviruses as intestinal pathogens in pigs [10]. PEDV was identified in over 50% of the diarrhea samples, which aligns with the significance of this virus for the global pig industry [10]. PEDV was first discovered in the UK in 1971, and subsequent, outbreaks occurred in various European countries and around the world [11]. The virus was initially isolated in Belgium and designated as CV777 [12,13]. Following this, countries such as China, South Korea, and Vietnam experienced multiple incursions of PED, resulting in a dramatic increase in piglet mortality and causing severe economic losses to the pig industry [14]. PEDV is also prevalent in North American pig populations, with cumulative economic impacts estimated to range from $900 million to $1.8 billion for the U.S. pig farming industry [15], including a recent estimate of $432 per sow [16]. The virus has since spread to several other countries, including Canada, Mexico, and Colombia [17]. The PEDV genome is approximately 28 kb in length and comprises 5′ and 3′ untranslated regions (UTRs) as well as multiple open reading frames (ORFs) [18]. It encodes several non-structural proteins (nsp1-nsp16, etc.) and four structural proteins: the spike protein (S), membrane protein (M), accessory membrane protein (E), and nucleocapsid protein (N) [19,20]. Research on the antigenic epitopes of PEDV primarily focuses on its structural proteins, with the S protein being the largest and having the most identified antigenic epitopes. The S protein serves as the main antigen that induces the production of neutralizing antibodies in the host, making it a primary target for the development of PEDV vaccines and therapeutics. The S protein consists of two subunits, S1 (1-789 aa) and S2 (790-1383 aa). S1 is the region that binds the virus to the host cell receptor (receptor-binding domain, RBD) and contains multiple neutralizing epitopes. Candidate vaccines based on S1 have demonstrated good immunogenicity in piglets.

TGEV was first identified in the United States in 1946, making it the earliest coronavirus detected in pigs [21]. The disease is currently prevalent in regions including the Americas, Asia, and Europe. TGEV can lead to enteritis, which may cause severe dehydration in newborn piglets, resulting in mortality rates that can reach up to 100% under unprotected conditions, particularly in piglets less than two weeks old [22,23]. The TGEV genome is approximately 28.5 kb in length, with a 5′-cap and a 3′-poly (A) tail structure at both ends [24]. Its open reading frames (ORFs) are arranged as follows: 5′-ORF1a-ORF1b-ORF2-ORF3a-ORF3b-ORF4-ORF5-ORF6-ORF7-3′ [24]. ORF2, ORF3, ORF4, and ORF7 encode four structural proteins: the spike protein, envelope protein, membrane protein, and nucleocapsid protein, which are crucial for viral assembly and immune evasion [23,24]. Among these, the N-terminal portion of the S protein (S1) is closely associated with TGEV’s recognition of target cells, the induction of neutralizing antibody responses, and the determination of the virus’s tissue tropism [25]. This region is located in the globular portion of the protein and is more exposed than the C-terminal part of the S protein. Studies have found that the S1 of TGEV contains four antigenic epitopes, which can induce a stronger immune response in mice compared to the full-length S gene [26,27].

Co-infection with PEDV and TGEV typically results in high morbidity and mortality rates among newborn piglets [9,28]. Vaccination has proven to be an effective strategy for preventing these infections, as supported by numerous studies [29,30]. Zhang et al. [2] developed the SL7207 DNA vaccine for TGEV and PEDV, which is delivered via attenuated *Salmonella* typhimurium, demonstrating its potential as an oral vaccine candidate for both diseases. Pascual-Iglesias et al. [31] engineered a PEDV-attenuated virus (rTGEV-RS-SPEDV) based on the TGEV genome, which effectively induces a PEDV-specific humoral immune response, as confirmed by experimental data. Currently, both inactivated and attenuated live vaccines for PEDV and TGEV are widely used. However, the emergence of highly virulent strains and repeated outbreaks, even on vaccinated farms, highlights the limitations of traditional vaccines, such as safety concerns and insufficient protective effects. Consequently, there is an urgent need to develop a safe and effective vaccine that can simultaneously prevent infections from both PEDV and TGEV. Therefore, we utilized a mouse model to evaluate PEDV S1 and TGEV S1 monovalent vaccines, a mixed vaccine of PEDV S1 and TGEV S1, and the PEDV S1-TGEV S1 combined vaccine, providing a potential method for preventing PEDV and TGEV infections.

## 2. Materials and Methods

### 2.1. Cells and Virus Culture

Vero cells (CCL-81) and the PEDV 2b strain YN15 were generously provided by Professor Qigai He at Huazhong Agricultural University. Swine testicular (ST) cells were obtained from BNCC (Henan, China). Both Vero cells and ST cells were cultured in Dulbecco’s modified Eagle’s medium (DMEM; Gibco, Grand Island, NY, USA), supplemented with 10% fetal bovine serum (FBS; Gibco, Grand Island, NY, USA), at 37 °C in a 5% CO_2_ atmosphere. The PEDV YN15 and TGEV HN2012 strains were propagated in Vero cells and ST cells, respectively, and the 50% tissue culture infective dose (TCID_50_) was determined using the Reed–Muench method [32].

### 2.2. Experimental Mice

Forty-eight 6-week-old female Kunming (KM) mice were purchased from Xinjiang Medical University in Xinjiang, China. All experimental procedures involving animals were approved by the Biology Ethics Committee of Shihezi University. The mice were provided with adequate food and clean water, maintained on a 12 h light–dark cycle, and kept at approximately 20 °C with 60% relative humidity.

### 2.3. Optimization and Synthesis of Genes for Expression and Purification of Antigens

The sequences of the PEDV S1 protein (GenBank accession number: QEM43340.2; amino acids (aa) 223-632), TGEV S1 protein (GenBank accession number: QCQ84262.1; aa 245-669), and its fusion protein (S1-S1) were codon-optimized to align with the codon usage preferences of *Escherichia coli* (*E. coli*). A His-tagged protein sequence was appended to the N-terminal for detection purposes, while a stop codon (TAA) was appended at the C-terminal. Additionally, restriction sites for *Nhe* I and *Hin*d III (Takara, Dalian, China) were included. The gene sequences were synthesized by Zoonbio Biotechnology (Nanjing, China), linked to pCZN-1 vectors, and subsequently transferred into the *E. coli* strain BL21 (DE3; Figure 1). The resulting recombinant vectors were designated as pCZN1-PEDV S1, pCZN1-TGEV S1, and pCZN1-PEDV S1-TGEV S1.

PEDV S1, TGEV S1, and PEDV S1-TGEV S1 proteins were expressed and purified as previously described [33]. Briefly, bacteria containing pCZN1-PEDV S1, pCZN1-TGEV S1, and pCZN1-PEDV S1-TGEV S1 were induced with IPTG (1 mmol/L; Solarbio, Beijing, China), collected, and resuspended in 25 mL of bacterial cell protein lysate. The cells were subjected to three cycles of freezing and thawing in liquid nitrogen and a 37 °C water bath, followed by ultrasonic disruption in an ice bath for 45 min. The mixture was then centrifuged at 12,000 rpm for 30 min to collect the precipitate. Proteins were purified according to the instructions provided with the His-Tagged Protein Purification Kit (CWBIO, Beijing, China).

### 2.4. SDS–PAGE and Western Blotting Assay

The expression and purification results for PEDV S1, TGEV S1, and PEDV S1-TGEV S1 proteins were validated as previously described [34,35]. In brief, the expression and purification of these proteins were confirmed through SDS–PAGE and Western blotting. An anti-His tag monoclonal antibody (diluted 1:4000; Solarbio, Beijing, China) was utilized as the primary antibody, while HRP-conjugated goat anti-mouse IgG (diluted 1:20,000; Solarbio, Beijing, China) served as the secondary antibody for immunoblotting analysis. The proteins were concentrated using ultrafiltration tubes (Millipore, Bedford, MA, USA), and their concentrations were determined following the instructions provided by the BCA protein quantification kit (Thermo Fisher Scientific, Waltham, MA, USA).

### 2.5. Vaccine Preparation and Animal Immunization

The quantified proteins were diluted to 1000 µg/mL and mixed with an equal volume of Montanide A206 water-in-oil adjuvant (SEPPIC, Courbevoie, France). This mixture was then emulsified using a high-shear dispersing emulsifier (FLUKO, Shanghai, China) at 4 °C for subsequent use. Forty-eight 6-week-old female KM mice were randomly divided into six groups and inoculated via intramuscular (IM) injection on days 0 and 14 (Table 1). The PBS immune group served as the negative control (NC), while the commercial vaccine group (Jilin Zhengye Biological Products Co., Ltd., Jilin, China) functioned as the positive control (PC). Following vaccination, the adverse effects in the mice were monitored in real time, and blood samples were collected at designated time points. The serum was then separated and temporarily stored in a refrigerator at −20 °C.

### 2.6. Enzyme-Linked Immunosorbent Assay

The levels of specific IgG, IgG1, and IgG2a antibodies in serum samples were quantified using an indirect enzyme-linked immunosorbent assay (ELISA), as previously described [36,37]. Briefly, the corresponding antigens were diluted according to preset protocols and added to 96-well ELISA plates (100 μL per well), followed by overnight incubation at 4 °C. After removing the coating solution, the wells were washed twice with PBST (Phosphate-buffered saline with Tween 20, Solarbio, Beijing, China) and dried. The wells were then blocked with 200 μL of 5% nonfat dry milk (BD, Franklin Lakes, NJ, USA) and incubated at 37 °C for 2 h, followed by two washes with PBST and drying. Diluted serum samples (100 μL) were added and incubated at 37 °C for 1 h. After five washes with PBST, 100 μL of HRP-conjugated goat anti-mouse IgG, IgG1, or IgG2a (Proteintech, Wuhan, China) was added and incubated for 1 h. Following five additional washes with PBST, 100 μL of TMB substrate (Solarbio, Beijing, China) was added and incubated for 15 min in the dark. The reaction was terminated by adding 50 μL of stop solution (Solarbio, Beijing, China), and the optical density (OD) at 450 nm was measured.

### 2.7. Determination of Neutralizing Antibody

Serum samples from mice were collected at 2, 4, and 6 weeks post-immunization to assess TGEV- and PEDV-specific virus-neutralizing (VN) activity. Briefly, heat-inactivated serum samples were serially diluted twofold (from 1:2 to 1:256) in DMEM medium, then mixed with an equal volume of TGEV or PEDV (200 TCID_50_/100 µL) and incubated at 37 °C for 1 h. Subsequently, 100 µL of the virus–serum mixture was added to a confluent monolayer of Vero or ST cells cultured in 96-well plates and incubated at 37 °C with 5% CO_2_ for 1 h. Finally, the mixtures were removed, the cells were washed twice with PBS, and maintained in 100 µL DMEM containing trypsin (10 µg/mL) for 3–5 days to observe TGEV- and PEDV-specific cytopathic effects (CPEs).

### 2.8. Enzyme-Linked Immuno-Spot (ELISPOT)

Interferon-gamma (IFN-γ) ELISPOT was used to evaluate the cellular immune response, as previously described [38,39]. Briefly, on day 28 post-prime immunization, three mice were randomly selected from each group, and their spleens were collected aseptically. The mouse spleen lymphocytes were separated using the mouse lymphocyte separation kit (TBD, Tianjin, China) according to the manufacturer’s instructions. The obtained lymphocytes were adjusted to 1 × 10^5^ cells/mL, resuspended in RPMI-1640 containing 10% FBS, and inoculated into an ELISPOT 96-well plate (200 μL/well) pre-coated with IFN-γ. Spleen lymphocytes were treated with corresponding proteins (10 μg) as the experimental group, while concanavalin A (ConA; 10 μg; Biosharp, Anhui, China) and PBS (Biosharp, Anhui, China) were used as positive and negative controls, respectively. Plates were incubated at 37 °C with 5% CO_2_ for 24 h in a cell culture incubator. IFN-γ spot-forming cells (SFCs) were then detected according to the manufacturer’s instructions (Mabtech, Nacka Strand, Sweden).

### 2.9. Statistical Analysis

Statistical analyses were conducted using GraphPad Prism 8.0.2 software (Graph-Pad Software Inc., La Jolla, CA, USA). A one-way analysis of variance (ANOVA) or a two-way ANOVA was employed to assess the differences among groups. All experiments were repeated at least three times. * *p*-values < 0.05 were deemed statistically significant.

## 3. Results

### 3.1. Construction of Recombinant Expression Vectors

To enhance antigen expression, the PEDV S1 and TGEV S1 genes were optimized, synthesized, and verified through double digestion with the *Nhe* I and *Hin*d III enzymes. The recombinant plasmids pCZN1-PEDV S1, pCZN1-TGEV S1, and pCZN1-PEDV S1-TGEV S1 were successfully constructed as confirmed by restriction enzyme digestion and sequencing (Figure 2).

### 3.2. Expression and Purification of Recombinant Proteins (Supplemantary Material Appendix A)

The supernatant and pellet obtained after ultrasonic disruption and centrifugation were subjected to SDS–PAGE analysis. The results showed that all three proteins were expressed as inclusion bodies, and high-purity proteins were obtained (Figure 3A). Western blotting analysis revealed specific immunoblot bands at approximately 41, 43, and 93 kDa, which were consistent with the expected results (Figure 3B).

### 3.3. Specific Antibody Levels Detection

To evaluate the humoral response induced by recombinant subunit vaccines in experimental animals, we immunized KM mice with subunit vaccines, using a commercial inactivated TGEV-PEDV vaccine as a PC and PBS as a NC. Serum samples were collected at 2, 4, 6, and 8 weeks post-immunization (wpi), and the levels of specific IgG, IgG1, and IgG2a antibodies were measured using the indirect ELISA method. The results demonstrated that both the commercial inactivated vaccine and subunit vaccines triggered strong induction of IgG, IgG1, and IgG2a in mice following immunization (Figure 4). By 8 wpi, the specific IgG antibody levels were significantly higher in the pCZN1-PEDV S1 + TGEV S1 immunized group compared to the PC (*p* < 0.001; Figure 4A). At 2 wpi, the specific IgG1 antibody levels were notably higher in the pCZN1-PEDV S1 + TGEV S1 immunized group than in the PC (*p* < 0.001; Figure 3B). By 4 wpi, all subunit vaccine groups exhibited significantly elevated levels of specific IgG1 antibodies compared to the PC (*p* < 0.0001; Figure 4B). Additionally, at 4 wpi, the specific IgG2a antibody levels in the pCZN1-PEDV S1 immunized group were significantly increased compared to those in the PC (*p* < 0.0001; Figure 4C).

### 3.4. Neutralizing Antibodies of PEDV and TGEV

To further evaluate the humoral response induced by subunit vaccines in experimental animals, we collected serum samples at 2, 4, and 6 wpi. The virus neutralization test (VNT) was subsequently employed to detect neutralizing antibodies against PEDV and TGEV. As illustrated in Figure 5A, the levels of PEDV neutralizing antibodies in the PEDV S1 and PEDV S1-TGEV S1 immunized groups were significantly higher than those in the NC at 4 wpi (*p* < 0.01). Furthermore, at 6 wpi, the levels of PEDV neutralizing antibodies in all vaccine-immunized groups were significantly elevated compared to the NC (*p* < 0.05). Notably, the level of PEDV neutralizing antibodies in the PEDV S1 + TGEV S1 immunized group was significantly higher than that in PC at 6 wpi (*p* < 0.05). As depicted in Figure 5B, the levels of TGEV neutralizing antibodies in all vaccine-immunized groups were significantly greater than those in the NC at both 4 and 6 wpi (*p* < 0.01). Additionally, the TGEV neutralizing antibody levels in the TGEV S1 immunized group were significantly higher than those in the PC at 2 and 4 wpi (*p* < 0.01), while the TGEV neutralizing antibody levels in the PEDV S1-TGEV S1 immunized group were significantly elevated compared to the PC at 4 wpi (*p* < 0.0001). These results suggest that subunit vaccines can elicit neutralizing antibody levels comparable to or exceeding those produced by inactivated vaccines.

### 3.5. The Effect of Recombinant Subunit Vaccines on Cytokine Expression

On the 28th day post-initial immunization, splenocytes were isolated and re-stimulated in vitro with the corresponding stimuli to analyze the cellular immune response. As shown in Figure 6, the level of IFN-γ produced by mice immunized with all subunit vaccines and the inactivated vaccine was significantly increased compared to the control group (*p* < 0.0001). The levels of IFN-γ produced by mice immunized with the PEDV S1, TGEV S1, and PEDV S1 + TGEV S1 vaccine groups were significantly lower compared to the PC (*p* < 0.001). However, the level of IFN-γ produced by mice immunized with the PEDV S1-TGEV S1 vaccine did not differ significantly from that of mice immunized with the PC (*p* > 0.05). These results suggest that all subunit vaccines can induce high levels of IFN-γ in mouse splenic lymphocytes, but only the PEDV S1-TGEV S1 vaccine induces levels of IFN-γ comparable to those of the inactivated vaccines.

## 4. Discussion

In recent decades, large-scale outbreaks of porcine diarrhea caused by PEDV and TGEV have occurred in the United States, Europe, and Asia, resulting in significant economic losses to the pig industry [9,40]. To date, vaccination remains one of the most effective measures to prevent outbreaks and epidemics of infectious diseases. Compared to traditional vaccines, subunit vaccines offer advantages such as safety, cost-effectiveness, efficiency, and ease of production [34,36]. The selection of appropriate foreign genes constitutes the first and crucial step in designing effective vaccines. The S1 protein, a structural domain of the S proteins of PEDV and TGEV, is located on the surface of the virus particles, possesses a high antigenic index, and can induce the production of neutralizing antibodies [2,41]. Therefore, we selected the S1 domains of the PEDV and TGEV S proteins as immunogens to develop effective vaccines aimed at preventing PEDV and TGEV infections. Currently, antibody-dependent enhancement (ADE) has been reported in West Nile Virus (WNV), Dengue Virus (DENV), Ebola Virus (EBOV), and coronavirus infections; however, we selected these two proteins with virus-neutralizing capabilities to mitigate the risk of ADE in vaccine development [42].

In this study, the S1 proteins of PEDV, TGEV, and the combined PEDV S1-TGEV S1 were purified, achieving purities exceeding 85%. The purified proteins were emulsified with A206 adjuvant in a 1:1 ratio, leading to the successful preparation of four water-in-oil subunit vaccines. KM mice were then immunized, and their immunogenicity was assessed. The results indicated that mice immunized with the subunit vaccines developed high levels of specific IgG, IgG1, and IgG2a antibodies. Notably, the levels of specific antibodies produced by some subunit vaccine groups surpassed those observed in the commercial vaccine group. For instance, at 8 wpi, the specific IgG antibody levels in the PEDV S1 + TGEV S1 immunized group were significantly higher than those in the commercial vaccine immunized group. Neutralizing antibodies, which directly reflect the protective capacity of the vaccine, are critical indicators for evaluating its immune protective effect [40]. Consequently, we also assessed the levels of neutralizing antibodies, finding that the subunit vaccines generated neutralizing antibody levels comparable to those of the commercial vaccine.

In addition to antibody responses, we also monitored the cellular immune responses in mice immunized with PEDV and TGEV subunit vaccines. IFN-γ, a cytokine produced by NK cells and T lymphocytes, enhances phagocytic activity and effectively eliminates pathogens. It has been reported that IFN-γ can induce a Th1 response by modulating chemotaxis and enhancing antigen presentation, thereby preventing pathogen infection [43,44]. Consequently, we measured the secretion levels of IFN-γ in the splenic lymphocytes of immunized mice. The results indicated that all subunit vaccines stimulated mouse splenic lymphocytes to produce high levels of IFN-γ. However, only the PEDV S1-TGEV S1 vaccine immunization group induced IFN-γ levels comparable to those in the commercial vaccine immunization group. Since mice are not susceptible to infection with PEDV or TGEV, we could not utilize this animal model for challenge experiments to assess the protective efficacy of the subunit vaccines. Nevertheless, our findings are consistent with earlier studies, highlighting the potential for developing highly effective subunit vaccines for the prevention and control of PEDV and TGEV [2,41]. These results enhance our confidence in evaluating the immune efficacy of subunit vaccines in pigs.

Currently, numerous researchers have developed new vaccines for PEDV and TGEV based on the S or S1 proteins using a variety of methods. These vaccines are essential tools for the prevention and control of PEDV and TGEV, as they can differentiate between vaccine-induced immunity and natural infection. However, these vaccines also present certain limitations, including limited or no protection against heterologous strains. Future studies should consider the development of a vaccine that incorporates the S1 gene from both the original and epidemic strains of PEDV, guided by epidemiological investigations, to provide more comprehensive protection.

## 5. Conclusions

In conclusion, we successfully prepared subunit vaccines for PEDV S1, TGEV S1, PEDV S1-TGEV S1, and PEDV S1 and TGEV S1, which induced robust cellular and humoral immune responses in mice. These findings lay a solid foundation for the development of safe and effective monovalent or bivalent vaccines against PEDV and TGEV. Furthermore, they strengthen our confidence in the next phase of evaluating the immune efficacy of these vaccines in pigs.

## Figures and Tables

**Figure 1 vetsci-12-00106-f001:**
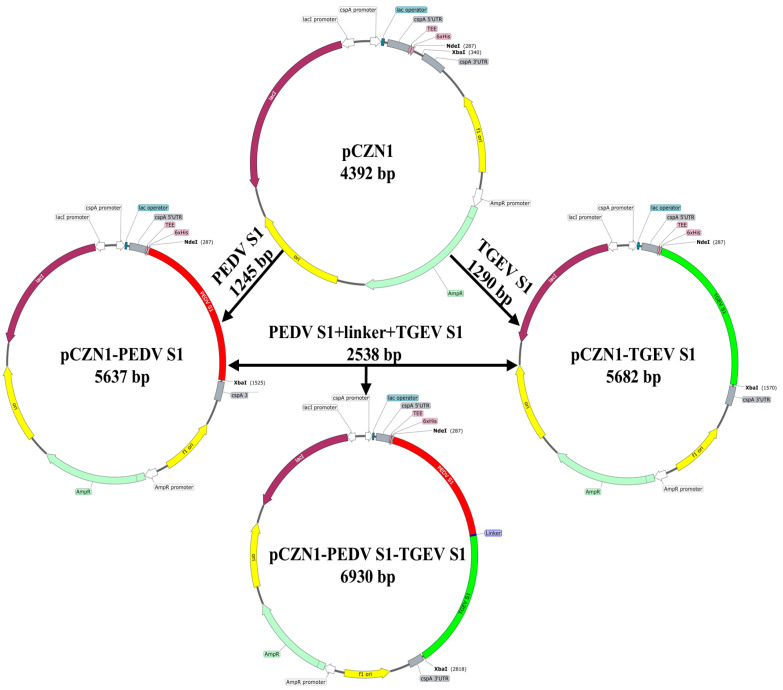
Schematic drawing of the construction of DNA plasmids. Plasmids pCZN1-PEDV S1, pCZN1-TGEV S1 and pCZN1-PEDV S1-TGEV S1 were generated according to the steps indicated by the arrows.

**Figure 2 vetsci-12-00106-f002:**
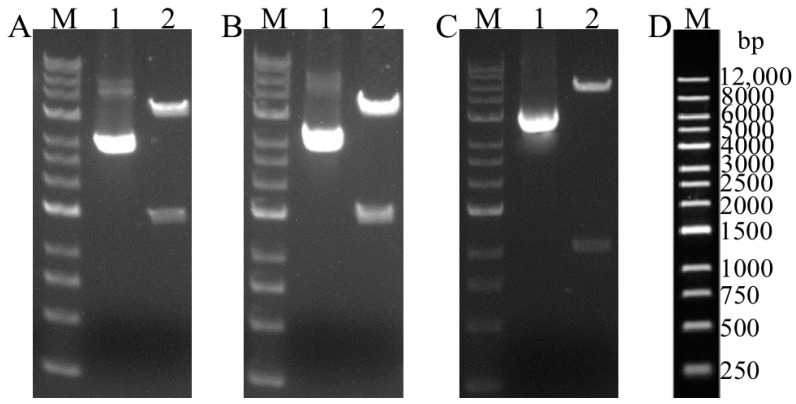
The result of the restriction enzyme digestion identification of the recombination plasmids pCZN1-PEDV S1, pCZN1-TGEV S1, and pCZN1-PEDV S1-TGEV S1. (**A**) Recombinant plasmid pCZN1-PEDV S1 double enzyme identification (*Nhe* I and *Xba* I). Lane M: 1 kb DNA Marker; Lane 1: pCZN1 plasmid digested; Lane 2: pCZN1-PEDV S1 recombinant plasmid digested. (**B**) Recom-binant plasmid pCZN1-TGEV double enzyme identification (*Nhe* I and *Xba* I). Lane M: 1 kb DNA Marker; Lane 1: pCZN1 plasmid digested; Lane 2: pCZN1-TGEV S1 recombinant plasmid digested. (**C**) Recombinant plasmid pCZN1-PEDV S1-TGEV S1 double enzyme identification (*Nhe* I and *Hin*d III). Lane M: 1 kb DNA Marker; Lane 1: pCZN1 plasmid digested; Lane 2: pCZN1-PEDV S1-TGEV S1 recombinant plasmid digested. (**D**) M:1 kb DNA Marker map.

**Figure 3 vetsci-12-00106-f003:**
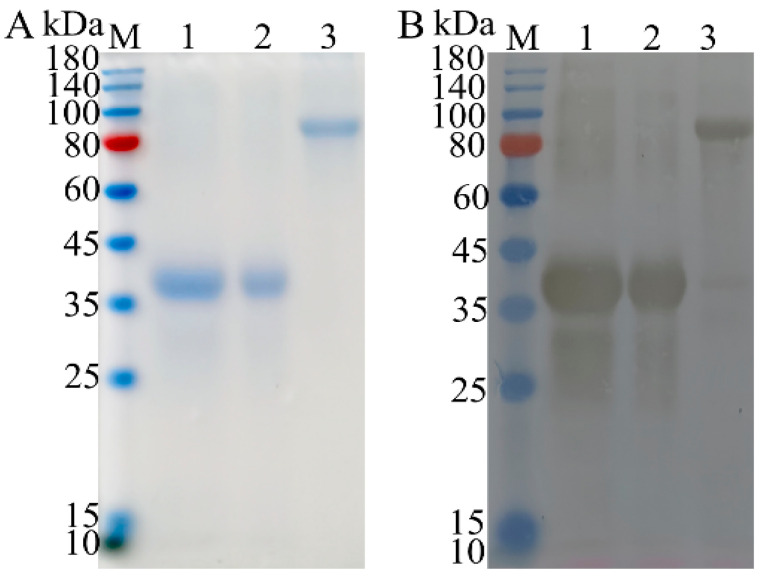
Expression and purification of recombinant proteins. (**A**) The expression of the PEDV S1, TGEV S1, and PEDV S1-TGEV S1 recombinant proteins was confirmed by SDS–PAGE with coomassie brilliant blue staining. Lane M: Protein marker; Lane 1: PEDV S1 recombinant protein; Lane 2: TGEV S1 recombinant protein; Lane 3: PEDV S1-TGEV S1 recombinant protein. (**B**) The expression of the PEDV S1, TGEV S1, and PEDV S1-TGEV S1 recombinant proteins was confirmed by Western blotting analysis using anti-PEDV S and anti-TGEV polyclonal antibodies. Lane M: Protein marker; Lane 1: PEDV S1 recombinant protein; Lane 2: TGEV S1 recombinant protein; Lane 3: PEDV S1-TGEV S1 recombinant protein.

**Figure 4 vetsci-12-00106-f004:**
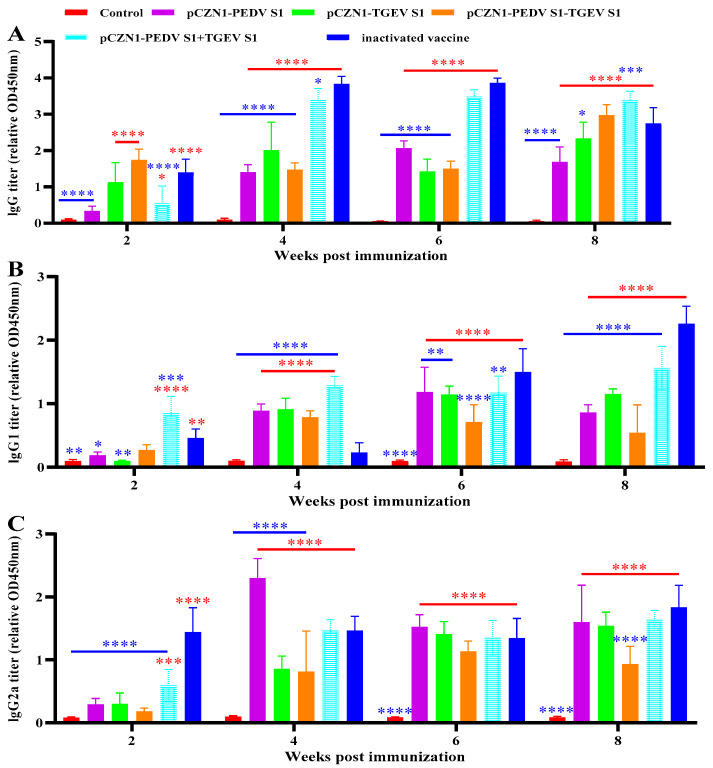
Specific antibody titers in immunized mice. KM mice were immunized with recombinant subunit vaccines, a commercial inactivated vaccine or PBS, respectively, and the serum was collected at 2, 4, 6, and 8 wpi, and the levels of specific IgG (**A**), IgG1 (**B**) and IgG2a (**C**) antibodies were detected by the indirect ELISA method. All performed experiments were repeated at least three times. * *p* < 0.05, ** *p* < 0.01, *** *p* < 0.001, and **** *p* < 0.0001, by two-way ANOVA. Note: * in red indicates comparison between subunit vaccines and the PC vs. the NC; * in blue indicates comparison between subunit vaccines and the NC vs. the PC.

**Figure 5 vetsci-12-00106-f005:**
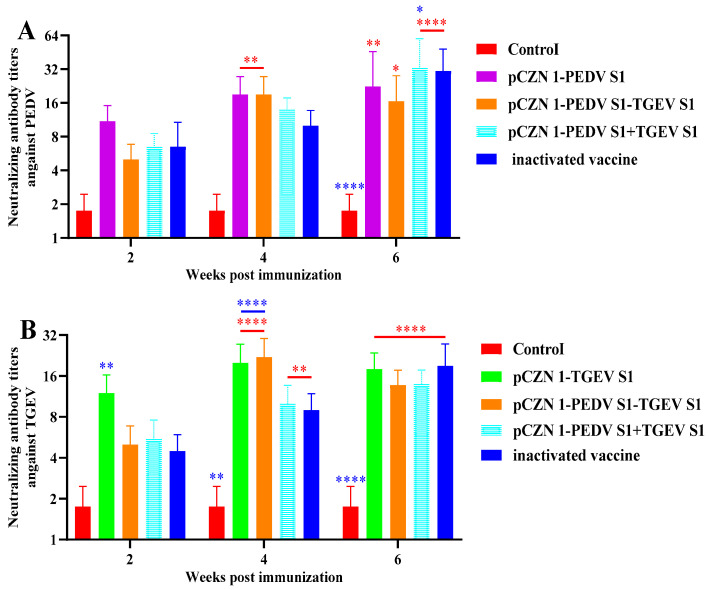
PEDV or TGEV-neutralizing antibody titers. All performed experiments were repeated at least three times. KM mice were immunized with subunit vaccines, a commercial inactivated vaccine, or PBS, and serum was collected at 2, 4, and 6 wpi for analysis of neutralizing antibody levels. (**A**) PEDV-neutralizing antibody titers. (**B**) TGEV-neutralizing antibody titers. * *p* < 0.05, ** *p* < 0.01, and **** *p* < 0.0001, by two-way ANOVA. Note: * in red indicates comparison between subunit vaccines and the PC vs. the NC; * in blue indicates comparison between subunit vaccines and the NC vs. the PC.

**Figure 6 vetsci-12-00106-f006:**
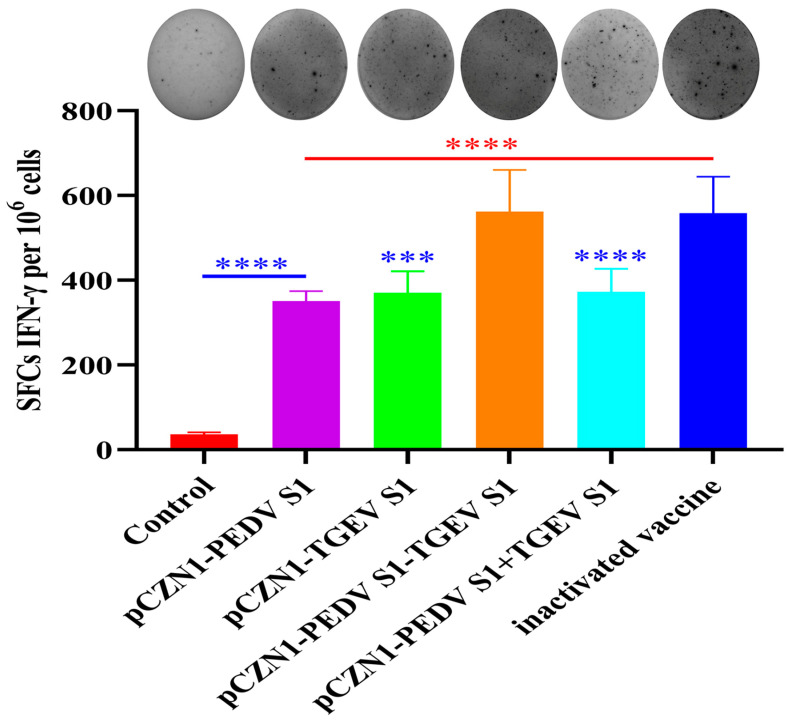
Immunization with subunit vaccines induced IFN-γ secretion. KM mice were immunized with subunit vaccines, a commercial inactivated vaccine, or PBS. Splenocytes were collected in the 28th day post-initial immunization, and re-stimulated with the same antigen used for immunization. The data were expressed as the number of IFN-γ secreting cells per 1 × 10^6^ splenocytes. The error bars represent the standard error of the mean. *** *p* < 0.001, and **** *p* < 0.0001, by one-way ANOVA. Note: * in red indicates comparison between subunit vaccines and the PC vs. the NC; * in blue indicates comparison between subunit vaccines and the NC vs. the PC.

**Table 1 vetsci-12-00106-t001:** Experimental grouping.

Groups (n = 8)	Immunogen and Dosage	Immunization Time (d)
PBS (Negative control, NC)	100 µL of Sterile PBS	0, 14th
PEDV S1 + A206 adjuvant	100 µg PEDV S1 + 100 µL A206 adjuvant	0, 14th
TGEV S1 + A206 adjuvant	100 µg TGEV S1 + 100 µL A206 adjuvant	0, 14th
PEDV S1-TGEV S1 + A206 adjuvant	100 µg PEDV S1- TGEV S1 + 100 µL A206 adjuvant	0, 14th
PEDV S1 + TGEV S1 + A206 adjuvant	50 µg PEDV S1 + 50 µg TGEV S1 + 100 µL A206 adjuvant	0, 14th
Inactivated vaccines (Positive control, PC)	200 µL	0, 14th

## Data Availability

The raw data can be obtained by contacting the corresponding author.

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
