# Peer review of "Development and Immunogenicity Study of Subunit Vaccines Based on Spike Proteins of Porcine Epidemic Diarrhea Virus and Porcine Transmissible Gastroenteritis Virus"

_vetsci, 2025, doi:10.3390/vetsci12020106_

Round 1
Reviewer 1 Report
Comments and Suggestions for Authors
The manuscript entitled "Development and Immunogenicity Study of Subunit Vaccines Based on Spike Proteins of Porcine Epidemic Diarrhea Virus and Porcine Transmissible Gastroenteritis Virus", in which the authors have constructed and characterized Spike proteins-based subunit vaccina against Porcine Epidemic Diarrhea Virus and Porcine Transmissible Gastroenteritis Virus. The study is interesting. However, it lacks novelty. In addition, It would have been more valuable if the study is performed in the natural host i.e. pig along with the virulent virus challenge model to determine the protective efficacy of the vaccine candidates. Further, the authors have used the E. coli protein expression system instead of eukaryotic expression system which lacks protein processing. With all the given limitations, the manuscript should be converted to short communication rather than full-length research manuscript before considering the manuscript for further review.
Minor comments:
Font size used in figure 1 is too small and not readable.
Comments on the Quality of English LanguageSeveral sections of the manuscript requires language editing.
Author Response
Firstly, we would like to express our sincere thanks for your kind letter and the constructive comments provided by the reviewers on our article (Manuscript Number: vetsci-3386539). These comments have been invaluable and have greatly contributed to the improvement of our manuscript. All authors have carefully considered each of the reviewers' suggestions. In response, we have made the necessary revisions to the manuscript in order to meet the requirements of your journal. Should any further revisions be necessary, we would be happy to address them and are grateful for your continued support. To facilitate the review process, we have highlighted the changes in the manuscript using two colors: red for content revisions and green for language improvements. A point-by-point response to the reviewers' comments is provided below.
|
(1) The manuscript entitled "Development and Immunogenicity Study of Subunit Vaccines Based on Spike Proteins of Porcine Epidemic Diarrhea Virus and Porcine Transmissible Gastroenteritis Virus", in which the authors have constructed and characterized Spike proteins-based subunit vaccina against Porcine Epidemic Diarrhea Virus and Porcine Transmissible Gastroenteritis Virus. The study is interesting. However, it lacks novelty. In addition, It would have been more valuable if the study is performed in the natural host i.e. pig along with the virulent virus challenge model to determine the protective efficacy of the vaccine candidates. Further, the authors have used the E. coli protein expression system instead of eukaryotic expression system which lacks protein processing. With all the given limitations, the manuscript should be converted to short communication rather than full-length research manuscript before considering the manuscript for further review. |
|
Response: Thank you very much for reviewing our manuscript and providing valuable suggestions, which have been truly inspiring. While we fully understand your suggestion to transition to a shorter communication, after careful discussion within our team, we have decided to retain the full research manuscript format. For vaccine construction, we chose the E. coli protein expression system because it offers a good balance of cost, operability, and potential for future optimization. It is cost-effective, scalable, and well-suited for the early stages of industrialization. Due to certain limitations, we were unable to perform validation in the natural host, pigs, which we deeply regret. However, we plan to collaborate with relevant institutions in the future to address this gap. (2) Font size used in figure 1 is too small and not readable. Response: Thank you for pointing this out. Your careful review has been invaluable in guiding the improvements made to the manuscript. Following your suggestion, we have enlarged the font in Figure 1 to ensure that readers can clearly view the diagram and understand the key information. (3) Several sections of the manuscript require language editing. Response: Thank you for your feedback. We have revised and optimized the manuscript to enhance its fluency and accuracy. The edited sections are highlighted in green. Please let us know if you have any further suggestions. |

Reviewer 2 Report
Comments and Suggestions for Authors
This study describes the production and immunogenicity in laboratory animals of a subunit vaccine for TGEV and PEDV. The work is well written, the sections well divided, and the methods used rigorous. Some flaws preclude its acceptance without revisions. Below are my specific comments:
1) Lines 50-52: It should be specified from the outset that these are results obtained in another study. Even in other introductory sentences, one should be more generic.
2) I advise the authors to cite in the introduction other works that have described co-exposure and co-positivity to TGEV and PEDV in pigs and wild boars in order to justify the use of a bivalent vaccine. Similar studies have been described in Campania (Italy). Another important study describes the possibilities of recombination: Coinfection and nonrandom recombination drive the evolution of swine enteric coronaviruses.
3) Introduction: Some more information about direct and indirect prophylaxis could be useful.
4) Merge "2.3" with antigen production.
5) More information about the ELISA should be specified.
6) Figure 4 presents important results but is poorly readable.
7) Different colors could be used in Figure 5, as some are similar and not easily distinguishable.
8) Figure 6 should be larger to be better appreciated.
Author Response
Firstly, we would like to express our sincere thanks for your kind letter and the constructive comments provided by the reviewers on our article (Manuscript Number: vetsci-3386539). These comments have been invaluable and have greatly contributed to the improvement of our manuscript. All authors have carefully considered each of the reviewers' suggestions. In response, we have made the necessary revisions to the manuscript in order to meet the requirements of your journal. Should any further revisions be necessary, we would be happy to address them and are grateful for your continued support. To facilitate the review process, we have highlighted the changes in the manuscript using two colors: red for content revisions and green for language improvements. A point-by-point response to the reviewers' comments is provided below.
(1) Lines 50-52: It should be specified from the outset that these are results obtained in another study. Even in other introductory sentences, one should be more generic.
Response: Thank you for pointing this out. Based on your feedback, we have made improvements.
Lines 50-52: A total of 127 porcine samples from 48 farms across six provinces in China were analyzed, revealing a PEDV detection rate of 43.0% and a 12.0% co-infection rate between PEDV and TGEV [5].
(2) I advise the authors to cite in the introduction other works that have described co-exposure and co-positivity to TGEV and PEDV in pigs and wild boars in order to justify the use of a bivalent vaccine. Similar studies have been described in Campania (Italy). Another important study describes the possibilities of recombination: Coinfection and nonrandom recombination drive the evolution of swine enteric coronaviruses.
Response: Thank you for pointing this out. Based on your feedback, we have included more references in the introduction that describe the co-exposure and co-infection of TGEV and PEDV in pigs and wild boars.
Lines 50-58: A total of 127 porcine samples from 48 farms across six provinces in China were analyzed, revealing a PEDV detection rate of 43.0% and a co-infection rate of 12.0% be-tween PEDV and TGEV [5]. Numerous studies have demonstrated that co-infection with PEDV and TGEV is prevalent, and may facilitate recombination between the two viruses [6-9]. Furthermore, evidence suggests that co-infection with these enteric vi-ruses can lead to synergistic or additive effects, resulting in more extensive villous atrophy and more severe, prolonged diarrhea throughout the entire intestine [3].
Relevant references (Lines 416-425)
- Huang, X.; Chen, J.; Yao, G.; Guo, Q.; Wang, J.; Liu, G. A TaqMan-probe-based multiplex real-time RT-qPCR for simultaneous detection of porcine enteric coronaviruses. Appl Microbiol Biotechnol. 2019, 103, 4943-4952. doi: 10.1007/s00253-019-09835-7.
- Zhou, J.; Wu, W.; Wang, D.; Wang, W.; Chang, X.; Li, Y.; Li, J.; Fan, B.; Zhou, J.; Guo, R.; Zhu, X.; Li, B. Development of a colloidal gold immunochromatographic strip for the simultaneous detection of porcine epidemic diarrhea virus and transmissible gastroenteritis virus. Front Microbiol. 2024,19,1418959. doi: 10.3389/fmicb.2024.
- Boniotti, M.; Papetti, A.; Lavazza, A.; Alborali, G.; Sozzi, E.; Chiapponi, C.; Faccini, S.; Bonilauri, P.; Cordioli, P.; Marthaler, D. Porcine epidemic diarrhea virus and discovery of a recombinant swine enteric coronavirus, Italy. Emerg Infect Dis. 2016, 22, 83-87, doi:10.3201/eid2201.150544.
- Guo, J.; Lai, Y.; Yang, Z.; Song, W.; Zhou, J.; Li, Z.; Su, W.; Xiao, S.; Fang, L. Coinfection and nonrandom recombination drive the evolution of swine enteric coronaviruses. Emerg Microbes Infect. 2024,13,2332653. doi: 10.1080/22221751.
(3) Introduction: Some more information about direct and indirect prophylaxis could be useful.
Response: Thank you for pointing this out. Based on your feedback, we have added the relevant information to the introduction.
Lines 107-113: Vaccination has proven to be an effective strategy for preventing these infections, as supported by numerous studies [30, 31]. Zhang et al. [2] developed the SL7207 DNA vaccine for TGEV and PEDV, which is delivered via attenuated Salmonella typhimurium, demonstrating its potential as an oral vaccine candidate for both diseases. Pascu-al-Iglesias et al. [32] engineered a PEDV-attenuated virus (rTGEV-RS-SPEDV) based on the TGEV genome, which effectively induces a PEDV-specific humoral immune response, as confirmed by experimental data.
Relevant references (Lines 471-479)
- Du, P.; Yan, Q.; Zhang, XA.; Zeng, W.; Xie, K.; Yuan, Z.; Liu, X.; Liu, X.; Zhang, L.; Wu, K.; Li, X.; Fan, S.; Zhao, M.; Chen, J. Virus-like particle vaccines with epitopes from porcine epidemic virus and transmissible gastroenteritis virus incorporated into self-assembling ADDomer platform provide clinical immune responses in piglets. Front Immunol. 2023, 14, 1251001. doi: 10.3389/fimmu.2023.1251001.
- Kong, F.; Jia, H.; Xiao, Q.; Fang, L.; Wang, Q. Prevention and control of swine enteric coronaviruses in China: a review of vaccine development and application. Vaccines (Basel). 2023, 12,11. doi: 10.3390/vaccines12010011.
- Pascual-Iglesias, A.; Sanchez, CM.; Penzes, Z.; Sola I.; Enjuanes, L.; Zuñiga, S. Recombinant chimeric transmissible gastroenteritis virus (TGEV)- porcine epidemic diarrhea virus (PEDV) virus provides protection against virulent PEDV. Viruses. 2019, 11,682. doi: 10.3390/v11080682.
(4) Merge "2.3" with antigen production.
Response: Thank you for pointing this out. This has provided us with a valuable perspective for optimizing our paper's structure. After careful discussion, we have adopted your recommendation and made the necessary revisions to this section.
Line 137:
2.3. Optimization and Synthesis of Genes for Expression and Purification of Antigens
(5) More information about the ELISA should be specified.
Response: Thank you for pointing this out. Based on your feedback, we have updated the manuscript to include the relevant details regarding the ELISA procedure.
Lines 185-198
The levels of specific IgG, IgG1, and IgG2a antibodies in serum samples were quantified using an indirect enzyme-linked immunosorbent assay (ELISA), as previously described [37, 38]. Briefly, the corresponding antigens were diluted according to preset protocols and added to 96-well ELISA plates (100 μL per well), followed by overnight incubation at 4 °C. After removing the coating solution, the wells were washed twice with PBST (Phosphate-buffered saline with Tween 20, Solarbio, China) and dried. The wells were then blocked with 200 μL of 5% nonfat dry milk (BD, USA) and incubated at 37 °C for 2 hours, followed by two washes with PBST and drying. Diluted serum samples (100 μL) were added and incubated at 37 °C for 1 hour. After five washes with PBST, 100 μL of HRP-conjugated goat anti-mouse IgG, IgG1, or IgG2a (Proteintech, China) was added and incubated for 1 hour. Following five additional washes with PBST, 100 μL of TMB substrate (Solarbio, China) was added and incubated for 15 minutes in the dark. The reaction was terminated by adding 50 μL of stop solution (Solarbio, China), and the optical density (OD) at 450 nm was measured.
(6) Figure 4 presents important results but is poorly readable.
Response: Thank you for pointing out the readability issue in Figure 4. Your careful review has been invaluable in guiding the improvements made to the manuscript. Based on your feedback, we have adjusted the chart type, font size, and color combination in Figure 4.
(7) Different colors could be used in Figure 5, as some are similar and not easily distinguishable.
Response: Thank you for pointing this out. Your careful review has been invaluable in guiding the improvements to the manuscript. Based on your feedback, we have adjusted the color combination in Figure 5.
(8) Figure 6 should be larger to be better appreciated.
Response: Thank you for pointing this out. Your careful review has been invaluable in guiding the improvements to the manuscript. Based on your feedback, we have enlarged Figure 6.
Round 2
Reviewer 1 Report
Comments and Suggestions for Authors
The authors have addressed all my comments.
Reviewer 2 Report
Comments and Suggestions for Authors
The authors have addressed all my previous comments, and the work is ready for your acceptance. I only recommend including further works in which co-infections and co-exposures to PEDV/TGEV are described in order to justify a possible vaccine (A Serological Investigation of Porcine Reproductive and Respiratory Syndrome and Three Coronaviruses in the Campania Region, Southern Italy; Retrospective Serosurvey of Three Porcine Coronaviruses among the Wild Boar (Sus scrofa) Population in the Campania Region of Italy)